# Multilingualism and Multiculturalism in *Family Guy*: Challenges in Dubbing and Subtitling L3 Varieties of Spanish

**Mariazell Eugènia Bosch Fábregas** [ID]

Department of Translation, Interpreting and Applied Languages, University of Vic-Central University of Catalonia, 08500 Vic, Spain; mariazelleugenia.bosch@uvic.cat

**Abstract:** Multilingualism and multiculturalism are verbally and visually recurrent in the sitcom *Family Guy* (1999-in production) through a combination of a main language of communication (L1) and other languages (L3) in the source language (SL) or source text (ST). The use of L3 is tantamount to tokenism and stereotyping characters, especially those whose recurrence is incidental and part of jokes. This paper compares two versions of the episode "Road to Rhode Island" (American and Spanish DVDs) and addresses a scene to analyze the linguistic challenges and lexical choices in dubbing and subtitling L1 and L3 in two geographical varieties of Spanish: Latin American Spanish and Peninsular Spanish. In this regard, this study focuses on the role and function of L3 in translation, the techniques to represent L3 in translation, L1 and L3 translation techniques, and which techniques are used in translation. Overall, this paper explores how the Spanish DVD adds a new L3 in the target text (TT) to maintain its original function in subtitling and dubbing, and the differences in the American DVD: $L3^{TT}$ omission in subtitling and $L3^{TT}$ change of function and meaning in dubbing, which ultimately accentuates linguistic and cultural misrepresentation and stereotypes.

**Keywords:** dubbing; DVD; *Family Guy*; L3; Spanish; stereotyping; subtitling

## 1. Introduction

American situational comedy *Family Guy* (1999–in production) is a mass media product that displays a great variety of values through visual and verbal content. It portrays a wide range of linguistic and cultural background characters that convey a multiplicity of socially and culturally controversial topics, namely homophobia, immigration, pedophilia, racism, religious tolerance and sexism. Such issues are framed by raw, aggressive, satirical, and controversial humor (LaChrystal 2012; Medina-Vicent 2012). *Family Guy*'s blatant satire and off-color jokes are materialized using "abrasive language and perceived indecent content" (LaChrystal 2012, p. 119), which can easily breach the limits of what an audience, and to a larger extent, societies, understand as morally permissible. In this line, "the repetitive use of derogatory speech is ritualized, and it could lead audience members to believe that it is acceptable and desirable" (pp. 121–22). *Family Guy*, as with any other series, is representative of a given time in a social and cultural context, so the interpretation of its values is greatly subjected to the place and time from which they are analyzed. Therefore, an active and conscientious viewing of the sitcom allows for transcending the humorous side of *Family Guy* and identifying serious issues that jokes cloak. The danger of not interpreting information not only lessens social and cultural importance, but it contributes to naturalizing and accepting values under the cliché it is just a joke.

Apart from humor, *Family Guy* is characterized by including a multiplicity of languages that internally encompasses the aforesaid topics. According to Rainier Grutman, the linguistic encounter of multiple languages represents multilingualism or the "co-presence of languages in a given society, text or individual" (Grutman 2019, p. 182). Since languages exist within cultures, multiculturalism must be considered to be an intrinsic part of multilingualism. During the last few decades, multilingualism has experienced rapid growth.

Christine Heiss (2004) documented a growing preference for culturally and linguistically diverse texts during the 1980s and 1990s. In more recent times, multilingualism and multiculturalism have been present in audiovisual content, thus making materials no longer and exclusively monolingual and monocultural (Meylaerts 2006; Heiss 2004).

Multilingualism or plurilingualism (Martínez-Sierra et al. 2010) becomes relevant in translation, since "every text is a collage of many texts in several languages" (Meylaerts 2010, p. 228) and cultures. For this reason, the translation of multilingual texts requires more than a linguistic transfer of a source text (ST) in a source language (SL) to a target text (TT) in a target language (TL). To put it simply, it involves "an interlinguistic and intercultural transfer of verbal and non-verbal information" (Chaume 2003, pp. 15–16). It must be noted that in *Family Guy*, translators face another constraint: a scripted language (Rey 2001) that shares similarities with natural-occurring conversation (Norrick and Spitz 2010; Quaglio 2009) and infuses language with realism (Beseghi 2019). Orality in *Family Guy* can be considered to be prefabricated (Chaume 2004, 2001) and the languages involved may be/sound natural or invented (Corrius 2008). However, multilingualism involves the inclusion of one or more languages in the format of L3, i.e., "a third language that refers to any other language(s) in the text" (Corrius and Zabalbeascoa 2011, p. 1) which "is distinct and distinguishable from the main language of any text, L1 in the ST and L2 in the TT" (Zabalbeascoa 2018, p. 165). In the original version of *Family Guy*, the main language of communication (American English as L1) can coexist with at least one L3 (henceforth L3$^{ST}$), such as Spanish, Italian, Chinese, etc. In the TT, L1 is translated (=L2), and so is L3 (henceforth, L3$^{TT}$). According to Zabalbeascoa (2018), for a third language to exist and be fully differentiated and understandable in both the ST and TT, it cannot be the same as L1 and L2.

*Family Guy* can be considered to be a perfect example of diversity due to the appearance and recurrence of characters that embody different cultural identities and linguistic backgrounds, "usually the butt of jokes that play on the language varieties and variations spoken by foreign people" (Iaia 2018, p. 154), such as Mickey McFinnigan, Peter Griffin's Irish dad, Consuela, the Latin American maid, Peter Griffin's boss Mr Weed, who is given a Hispanic accent, Swedish characters Tomik Björk and Bellgarde as the "two foreign guys [ . . . ] who've been living in the US almost long enough to sound American"[1] or even the British characters Nigel Pinchley and James William Bottomtooth III. However, a biased portrayal of cultures and languages in the sitcom distinguishes two groups: American English-speaking white characters—mainstream culture—and the stereotypical representations of the Other (Said 1978). In other words, the series portrays a blend between a main language of communication and "other languages". In this line, Di Giovanni refers to cultural otherness as "the depiction of cultures which are distant in space or time from the familiar cultural background" (Di Giovanni 2003, p. 208). In the series, alterity is often epitomized by migrant characters (i.e., Asians, Latin Americans, Italians, among others) whose command of their native language(s), such as Chinese, Spanish, Italian, Korean, etc., is excellent while their competence and performance of American English is limited. In fact, "multilingual written or audiovisual texts do not give equal prominence to the languages they display" (Grutman 2019, p. 342). In *Family Guy*, the use of L3 appears in very specific situations, often briefly but always denoting given connotations and carrying out definite purposes for the main plot or scene(s). In such cases, poor linguistic proficiency in English often influences L3 characters' mother tongue(s), a fact that suffices to label their English performance as "broken English", which ultimately stereotypes migrant characters as unintelligent and illiterate.

Categorization contributes to creating and naturalizing differences based on language and culture. In *Family Guy*, difference is not only perceived through non-white characters speaking their native languages but also in the way they are employed to represent cultural and linguistic status based on inferiority. In fact, other languages and cultures are often relegated to comical purposes and stereotyping. Characters representing otherness in *Family Guy* are either incidental or part of jokes, i.e., their appearance in the series is limited,

they are not essential for the main plot, and hardly ever is their presence continued in other episodes. Another common feature of these characters is their lack of a name. Anonymity can result in invisibility, but also in stereotyping. Characters are consequently pigeonholed, reduced, and distorted, which ultimately accentuates social and cultural misrepresentations based on class, race, sex, and purchasing power under a humorous perspective (Sokoli et al. 2019; Medina-Vicent 2012; Bonaut and García-López 2010). The major portrayal of otherness in *Family Guy* is characterized by the Latin American community, which is described as a "fusion of all Latin Americans that emigrate to the United States under the same term: Mexican" (Medina-Vicent 2012, p. 540). Since the use and representation of other languages in *Family Guy* is tantamount to tokenism and stereotyping of characters, communities, and cultures, this paper analyzes the biased representation of the Latin American community and aims at giving visibility to a nameless character whose linguistic and cultural background differs from the mainstream language and culture (L1).

This situation is particularly relevant in cases where American English (L1) is mixed with geographical varieties of Spanish (L3), namely Latin American Spanish and Peninsular Spanish, in translation (L2 and L3$^{TT}$). Multilingualism and/in translation have an unarguably intrinsic relationship. In fact, "at the heart of multilingualism, we find translation" (Meylaerts 2010, p. 227). Contexts based on the plurality of languages and cultures have sparked research on the role of different languages in/and translation, but no research has addressed the translation of multilingual and multicultural environments in *Family Guy*. For this reason, this study analyzes and compares the linguistic challenges and lexical choices in dubbing and subtitling L1 and L3 in two varieties of Spanish in one scene of an episode of *Family Guy* to (1) check possible differences in L3′s role and function in translation (Corrius and Zabalbeascoa 2011; Corrius 2008) (2) identify the different techniques to represent and mark L3$^{TT}$ (Bartoll 2006), L1 and L3 translation techniques (Martínez-Sierra et al. 2010), and the type of techniques used in translation (Molina and Hurtado 2002), (3) test the extent to which Latin American and Peninsular Spanish reinforce and/or perpetuate linguistic and cultural misrepresentation and stereotypes and (4) give voice to an unknown but representative character of a multilingual and multicultural environment.

## 2. Materials and Methods

The study of L1 and L3 in translation through Latin American Spanish and Peninsular Spanish is analyzed in the *Family Guy* episode "Road to Rhode Island". Since two varieties of Spanish are taken into consideration, it is necessary to compare the American DVD of the series (*Family Guy*) released in 2003, and the Spanish DVD of the sitcom (*Padre de Familia*), released in 2005. The reason for choosing the DVDs of the series is because they provide "an original version with subtitles in the target language of the particular country [they are] sold (sometimes also with subtitles in the source language) and a dubbed version (with subtitles for the deaf on demand)" (Heiss 2004, p. 216). In this regard, DVDs offer the official interlingual and intralingual subtitles. In the American (US) DVD, "Road to Rhode Island" becomes "Camino a Rhode Island" in Latin American Spanish and it corresponds to the sixth episode of the second season (S2E06). In the Spanish DVD, the Spanish title of the episode is "El Camino a Rhode Island". It is also included in the second season, but it corresponds to the thirteenth episode[2]. As mentioned previously, this paper focuses on a short scene (Timecode 12:19′–12:50′ in the US DVD and 12:37′–13:04′ in the Spanish DVD) that is loaded with multiculturalism, multilingualism, stereotyping, and humor both in the subtitling and dubbing versions. Despite its shortness, it is uncommon to find an episode in *Family Guy* that combines the above-mentioned issues and that they become significantly relevant in translation.

In "Road to Rhode Island", the Griffin's pet, Brian, sets on a quest from the fictional town of Quahog (Rhode Island) to Austin (TX, USA) to find his long-lost mother. In the scene for this analysis, he travels with Stewie in a wooden trailer with some passengers. The video images portray a crowd of non-white characters wearing hats and unkempt clothes, an image that encapsulates the tendency of conveying a stereotypical version of

Latin American immigrants in *Family Guy* (Medina-Vicent 2012). Brian feels unsure as to whether they are traveling in the right direction, so he approaches one of the passengers and asks for directions. In the English version of both the American and Spanish DVDs, Brian does not use the main language of communication (L1), i.e., his native tongue (American English). Instead, he addresses the man in Spanish (L3$^{ST}$). The reason he employs Spanish is probably reinforced by his misconception of Latin Americans not having a good command of the English language, a fact that is oftentimes expressed through complete ignorance of English or the use of broken English (Medina-Vicent 2012). To his surprise, the passenger knows both Spanish and English. In this regard, the man can follow the conversation in both languages and correct Brian's ungrammatical use of Spanish in English, therefore proving the inconsistency of linguistic stereotypes applied to the Latin American community. Overall, the exchange of information becomes a metalinguistic discussion and a linguistic confusion between English and Spanish that results in both characters understanding and misunderstanding each other at the same time.

Defining a multilingual text as one that is "worded in different languages" (Delabastita and Grutman 2005, p. 15) is incomplete unless the concept of "language" and "language variety" (De Heredia and De Higes 2019; Delabastita and Grutman 2005) is addressed. In fact, multilingualism encompasses "a standard variety of the language with standard varieties of that same language in other territories, with nonstandard dialects or with other languages (invented or not), with jargon and with different registers of language" (De Heredia and De Higes 2019, p. 12). As for L3, which has previously been introduced, Corrius and Zabalbeascoa claim that it is

> the defining feature of multilingual source texts as well as translations, and it is either a distinct, independent language or an instance of relevant language variation to highlight the presence of more than one speech community being portrayed or represented. (Corrius and Zabalbeascoa 2011, p. 115)

Though it may seem that multilingual texts only contain one L3, in fact, "a seemingly ever-increasing number of texts, written or audiovisual are not restricted to a single language or a single standard variety" (Zabalbeascoa 2018, p. 165). In *Family Guy*, there are cases in which "a given text [ . . . ] might contain [ . . . ] one or more "token languages (L3a + L3b)" (165; also, Zabalbeascoa and Voellmer 2014), as shown in this study. As mentioned, this paper tackles multilingualism and multiculturalism in translation through the role and function of L3 dubbing and subtitling of a scene in two geographical varieties of Spanish (i.e., Latin American and Peninsular Spanish), together with the techniques that have been used to convey L1 and L3 in translation. Depending on how L1 and L3 have been dubbed and subtitled (L2 and L3$^{TT}$) in each DVD, a possible situation of linguistic similarity (L2 ≈ L3$^{TT}$) or equivalence (L2 = L3$^{TT}$) can result in overlap, a change of L3$^{TT}$ function or even a L3$^{TT}$ loss (i.e., the loss of difference conveyed by L3$^{ST}$). Before addressing L3 in translation, Corrius and Zabalbeascoa (2011) advise translators to identify certain factors in relation to L3$^{ST}$, i.e., (1) L3 presence (spoken or written, as in captions, credits, inserts, subtitles) and (2) audience familiarity with L3 language and culture.

Based on the previous criteria, Corrius and Zabalbeascoa (2011) suggest different operations to translate L3$^{ST}$ segments and the possible results and effects of such techniques. In this regard, translators might delete L3, but a loss of L3$^{TT}$ can result in standardization. In situations where L3$^{ST}$ differs from L2 (L3$^{ST}$ ≠ L2), L3$^{ST}$ and L3$^{TT}$ can be repeated. Since L3$^{TT}$ will equal L3$^{ST}$, L3 in the target text will be kept, even though the function and connotation may change. Translators can substitute L3$^{ST}$ and L2 in cases where L3$^{ST}$ is different from L2 (L3$^{ST}$ ≠ L2). When L3$^{ST}$ is the same as L2 "and it is left unchanged, it may lose its visibility and L3 quality" (Zabalbeascoa 2018, p. 176). In both cases, L3$^{TT}$ segment is lost, which can culminate in L3 invisibility, and standardization (with or without compensation) unless some L2 strategy is applied, such as "*signalization*, the literal naming of a language in the text" (Bleichenbacher 2008, p. 59). Finally, L3$^{ST}$ can be substituted when L3$^{ST}$ is different or equal to L2 (L3$^{ST}$ ≠ L2 or L3$^{ST}$ = L2). In both cases, L3 in the TT is kept because it can differ from L3$^{ST}$ and L2 (L3 ≠ L3$^{ST}$/L2) and it can equal L1 or

not ($L3^{TT} \neq /= L1$). As a result, both operations can convey an equivalent or analogous function or connotation. Overall, L3 in the target text can be classified in terms of its degree of manipulation: "*unchanged* ($L3^{TT} = L3^{ST}$), *neutralized* ($L3^{ST} = L2$), since it cannot be differentiated in the TT or it has been deleted, and *adapted* ($L3^{TT} \neq L3^{ST}$), implying that nor has it been neutralized or omitted" (Corrius 2008, p. 383), which indicates that "L3 has been substituted for a language that is different to L2" (Corrius and Zabalbeascoa 2011, p. 120).

As mentioned earlier, the study on L3 role and function also considers the different techniques used to represent L3 in translation. For this, the analysis focuses on Bartoll's (2006) research and the strategies employed in multilingual films as regards to (1) marking $L3^{TT}$ by means of italics/colors ($L3^{TT} + L2$), intralinguistic subtitling ($L3^{TT} + L3^{TT}$), no translation and no translation indicating language in brackets ($L3^{TT}$) or (2) not marking L3: normal font ($L3 + L2$). Together with Bartoll (2006), L1 and L3 will be examined taking into consideration the translation techniques in the original version (Martínez-Sierra et al. 2010). Finally, the paper also revisits which translation techniques (Molina and Hurtado 2002) have been used in dubbing and subtitling L3. As Molina and Hurtado clarify, "strategies and techniques occupy different places in problem-solving: strategies are part of the process, techniques affect the result" (508). Since this paper tackles the resulting L3 role and function in translation, it is of great importance to choose translation techniques (and not strategies) as the "procedures to analyze and classify how translation equivalence works" (Molina and Hurtado 2002, p. 509). In this regard, this study focuses on techniques such as adaptation, amplification, borrowing, calque, compensation, etc. used in translation to identify the extent to which the translators left any personal marks (De Marco 2012).

## 3. Results

A comparison between the American (US) and the Spanish (SP) DVD has revealed a different use of languages when it comes to the main language of communication (L1) and other languages (L3) in both dubbing and subtitling versions. In the original audio and intralingual subtitles of both DVDs, L1 is American English. Due to differences in region, L3 changes in the American and Spanish DVDs. For this reason, L3 in the source text (ST) audio is Latin American Spanish on the American DVD and Peninsular Spanish on the Spanish DVD. As for subtitling $L3^{ST}$, Peninsular Spanish can be read in the Spanish DVD. In the American DVD, however, $L3^{ST}$ does not provide subtitles. Instead, the caption "[Speaking Spanish]" appears onscreen when Brian speaks Latin American Spanish. This means that the audience can hear that he speaks another language, but they cannot read the subtitles. The conversation is transcribed in Table 1 (American English audio and dubbing in the two varieties of Spanish) and Table 2 (American English intralingual subtitling and Latin American and Peninsular Spanish interlingual subtitling).

**Table 1.** Transcription of the English audio, Latin American, and Peninsular Spanish dubbing.

| Speaker | American English Audio | Latin American Dubbing | Peninsular Spanish Dubbing |
|---|---|---|---|
| Brian | Hola. Ah, me, me llamo es Brian. Ah, let's see. Nosotros queremos ir con ustedes. | Hola, mi . . . mi nombre es Brian. Y quiero, quiero . . . No, no, eh . . . queremos, nosotros queremos ir con ustedes. | Hello. My name is de Brian. Ah, veamos. We have . . . we have to go with you. |
| Traveler | That was pretty good. But actually, when you said "Me llamo es Brian", you don't need the "es". | Ah, eso está muy bien. Pero déjeme decirle que te encontraste con la persona equivocada. | Eso ha estado muy bien. Pero cuando dice "*My name is de Brian*", no diga "*de*". |
| Brian Traveler | -Just "Me llamo Brian". -Oh, you speak English. | Yo no soy ningún pollero. ¿No eres pollero? | -Sólo "*My name is Brian*". -Oh, habla mi idioma. |
| Traveler Brian | -No, just that first speech and this one explaining it. -You, you're kidding, right? | No, soy un simple ilegal. Pero, ¿bromeas, cierto? | -No, yo hablo inglés, pero es que estoy doblado. -¿Me toma el pelo? |

**Table 1.** *Cont.*

| Speaker | American English Audio | Latin American Dubbing | Peninsular Spanish Dubbing |
|---|---|---|---|
| Traveler | ¿Qué? | No. | *What?* |
| | | Austin, doce kilómetros | |
| Brian | ¡Señor, pare el auto! | ¡Oiga, deténgase! | *¡Señor, pare el carro!* |

**Table 2.** Transcription of English (US; SP DVD), Latin American, and Peninsular Spanish subtitling.

| Speaker | Eng. Sub. (US DVD) | Eng. Sub. (SP DVD) | Lat. Am. Spanish Subtitling | Peninsular Spanish Subtitling |
|---|---|---|---|---|
| Brian | [Speaking Spanish] | Hola. Me llamo es Brian. | | *Hello. My name is de Brian.* |
| | [ … ] | [ … ] | [ … ] | [ … ] |
| Traveler | But when you said, "*Me llamo es Brian*," you don't need the "*es*". | But when you said "Me llamo es Brian", you don't need the "es". | Pero cuando dices "Me llamo es Brian", no necesitas el "es". | Pero cuando dice "*My name is de Brian*", no diga "*de*". |
| Brian | You speak English. | Oh, you speak English. -No just that first speech and this one. | Hablas inglés. | -Habla mi idioma. |
| Traveler | Just that first speech. And this one explaining it. | | Sólo eso. Y esto que estoy diciendo. | -No, hablo inglés, pero estoy doblado. |
| Brian | -You're kidding, right? | -You're kidding, right? Qué? | | -¿Me toma el pelo? |
| Traveler Brian | *-Qué?* [Speaking Spanish] | Señor, pare el auto! | -Bromeas, ¿verdad? | *¿What?* *¡Señor, pare el carro!* |

In the American English audio and subtitling of the US and SP DVDs, Brian's stereotyped image of Latin Americans probably led him to assume that the traveler (a) did not speak English, (b) spoke "broken English" and (c) knew and spoke Spanish. As a result, Brian struggles to communicate with the man in Spanish, since it is not his mother tongue: "Hola. Ah, me, me llamo es Brian" (lit. "Hello. Hm, I, I am called is Brian"). Brian's lack of Spanish knowledge makes him hesitate ("ah", "let's see") during his speech, repeat structures ("me, me"), and make grammatical mistakes. After introducing himself, Brian continues speaking in Spanish (this time using a correct structure): he manifests his intention of traveling together to seek confirmation as to whether he goes in the right direction. The man seems to have understood Brian's Spanish from the beginning, but he does not provide an answer to his request. Instead, he praises and corrects Brian's performance of Spanish in English. After the metalinguistic explanation, Brian is puzzled at realizing that the man does speak his language (American English). The traveler, however, clarifies that his knowledge of English is reduced to "the first speech and this one explaining it". Brian thinks that he is being teased, so when he confronts the man, the latter asks for clarification in "¿Qué?" (lit. "What?"). It is unclear whether the traveler purposefully misleads Brian for comical purposes or whether he is competent in both English and Spanish (or both situations). However, in positioning himself as a more competent speaker and the fact that he has corrected Brian, the passenger is metaphorically placed in a higher linguistic position than Brian. This reasoning could be a slight attempt to mitigate the stereotype of migrants having "slow and hesitant speech, English-language difficulties and little ability to express and understand others" (Medina-Vicent 2012, p. 542), which characterizes the Latin American community in *Family Guy*.

*3.1. Dubbing L3$^{ST}$: Latin American Spanish and Peninsular Spanish*

As regards the Latin American and Peninsular Spanish dubbed versions (Table 1), there are some striking differences in the interchange of information, meaning, and purpose of the conversation and ultimately, the role and function of L3 in the target text (TT), as discussed in the analysis below.

Peninsular Spanish dubbing is very similar to American audio and subtitling in the sense that English and one geographical variety of Spanish are used. However, Peninsular Spanish dubbing presents a linguistic role reversal. In the English audio and subtitling of both the American (US) and Spanish (SP) DVDs, Brian tries to speak Latin American Spanish (L3 in the source text) while his native language is English (L1). In Peninsular Spanish dubbing, however, he attempts to speak English (L3$^{TT}$) by assuming that his fellow traveler is foreign and does not speak Brian's native language, i.e., Spanish (L2). The fact that L3$^{TT}$ is English in Peninsular dubbing might be sufficient to consider that the passenger only knows/speaks English. Nevertheless, dubbing reveals that he knows English and he speaks Peninsular Spanish, and not Latin American Spanish. In the English audio (and Latin American dubbing), however, the Spanish variety corresponds to a Latin American Spanish accent. This is noticed through the different tone between Peninsular Spanish and Latin American Spanish, the recurrent treatment of formal "usted" (lit. you) and "seseo", i.e., the pronunciation of the sound "s" for letters "c" and "z" when placed between vowels, as in cases such as "diciendo" (lit. "saying").

Despite the linguistic reversal, Peninsular Spanish dubbing successfully maintains the linguistic confusion that appeared in the English audio and subtitling. In this case, however, Brian must have assumed that the other person did not speak Peninsular Spanish but English, which is plausible since they are both in North America. The reason behind such linguistic inversion in SP dubbing avoids confusion in translation. If L3$^{ST}$ had been rendered to Latin American and L1 to Peninsular Spanish, Brian and the man would have understood each other due to linguistic similarity between Peninsular and Latin American Spanish (L2 ≈ L3$^{TT}$). Consequently, L3$^{TT}$ would have been neutralized. In none of the English versions does Brian speak Latin American Spanish fluently, so in Peninsular Spanish dubbing he resorts to a poorly "Hello, my name is de Brian". Moreover, he stutters before producing an unsure and hesitating English use of "We have . . . we have . . . ". Like the English audio and subtitling, Peninsular Spanish dubbing also shows some grammatical mistakes, which convey the idea of Brian not having a good command of English. The mistake in "my name is de Brian" could be explained due to the existing homophony between the Spanish preposition "de" (lit. "of") and the English definite article "the". As a result, Brian conveys a non-grammatical structure that sounds like "my name is the Brian". After Brian's performance is corrected, he is pleased to know that his fellow traveler speaks the same language: "habla mi idioma" (lit. "you speak my language").

Interestingly, Brian does not specify whether the man speaks English or Peninsular Spanish. Instead, he resorts to the hypernym "idioma" to avoid language precision, which is a common convention in dubbing. In fact, Brian is indirectly referring to Peninsular Spanish, as it is the language that he is speaking. Moreover, the accent in dubbing confirms it. However, since he is traveling to Mexico, he could have spoken Latin American Spanish. Had it been the case, Brian and the traveler would have understood each other perfectly. In this line, Brian's unspecified use of "mi idioma" indicates that they speak a similar language, regardless of it being Peninsular Spanish or Latin American Spanish. The traveler immediately clarifies that he does not speak the language: "No, yo hablo inglés, pero estoy doblado" (lit. "I speak English, but I am dubbed").

The difference, however, is that in Peninsular Spanish dubbing, the man does not deny speaking any geographical variety of Spanish (he does speak, since he is dubbed), but simply argues that he speaks English. In the Spanish DVD, dubbing is not only interesting regarding the ambiguous use of "mi idioma" but also because there is a case of metalinguistic wordplay, which to some extent could be considered to be a "metatranslative" wordplay, as explained below. Comparing the American English audio and subtitling to

the Peninsular Spanish dubbing, the inclusion of "doblado" is relevant for several reasons. First, the word did not appear in the source text/language, i.e., in any of the American English versions. Instead, it corresponds to a humorous and intentional note, a personal mark (De Marco 2012) that the translator purposefully introduced, or even was proposed in the dubbing studio, probably by the dubbing director.

Whatever the case, a source text/language context without a pun in the American English audio/subtitling is rendered to a pun in the target text (Henry [1993] 2003; Delabastita 1994, 1996; Heibert 1993). In terms of Delabastita (1994), for a specific pun to work in the context, its "intention, function, effect, and communicative value" (223) need to be considered. As for "doblado", the pun works as a polysemic word because it can be interpreted in two different ways. On the one hand, it refers to a curved physical position, i.e., "bent". Guided by the visual support of the scene (Figure 1), the sitcom represents the traveler on the floor, tuck sitting. On the other hand, "doblado" refers to how a language has been rendered, in other words, "dubbed". In this line, the traveler knew English, but he had been adapted into Spanish, in other words, he has been "doblado". Overall, the choice of this word included a witty pun whose purposeful comical touch reminds the audience of the strong connection between verbal and visual language. In this case, the target language allowed wordplay that incremented the humorous tone of the scene without changing meaning.

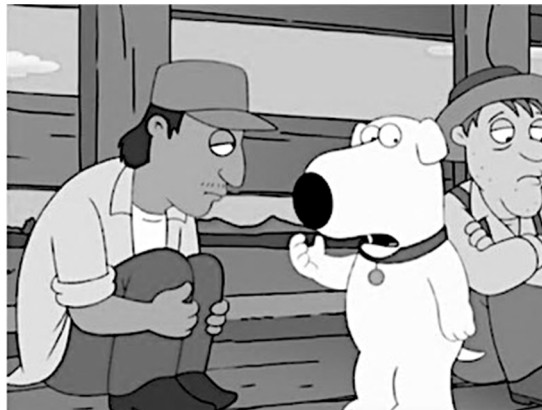

**Figure 1.** "Doblado" (Screenshot) Source: Spanish DVD (2005).

So far, the main difference between English audio/subtitling and Peninsular Spanish dubbing is that American English (L1) becomes Peninsular Spanish (L2) in translation (SP DVD), and that Latin American Spanish (L3$^{ST}$) is rendered as American English (L3$^{TT}$) in Peninsular Spanish (SP DVD). However, the American accent in the Spanish DVD is not the same as in the American DVD, and Brian's accent becomes Hispanicized. The language reversal in the TT allows maintaining the linguistic difference between L2 (Peninsular Spanish) and L3$^{TT}$ (American English), in other words, L2 ≠ L3, but it also preserves L3$^{ST}$ function in translation. As mentioned previously, the Spanish DVD is commercialized in Spain, and because of this, producers might have decided not to dub L3$^{ST}$ in Latin American Spanish to avoid confusion between languages, not further stereotype the character, etc.

Brian ends the conversation when he sees a road sign indicating Austin. At that moment, he addresses the driver and instructs him to stop the vehicle: "¡Señor, pare el carro!" (lit. Sir, stop the car!). In the American English audio and subtitling, Brian uses the word "auto", which is another form of "automobile" or "car" (DRAE 2023; María Moliner 2000, p. 145). In Peninsular Spanish, "carro" may refer to a "carriage" (often a two-wheel pulled vehicle) or a "trolley" (DRAE 2023). In Latin American Spanish, "carro" means "car" or a "train wagon where one can sleep" (María Moliner 2000, p. 255). In the context of the scene, characters are probably going to Mexico, and they are likely to have been traveling for days and nights, so Moliner's definition of "train wagon" apply. In Peninsular Spanish dubbing, therefore, Brian's use of "carro" as a single regional word not only suggests an

instance of signalization (Bleichenbacher 2008), i.e., the introduction of Latin American Spanish (another L3), but it purposefully differentiates the two geographical varieties of Spanish, thus indicating that while Brian speaks Peninsular Spanish, he is in a context where Latin American Spanish is spoken.

A quick comparison between Latin American Spanish and Peninsular Spanish dubbed versions (Table 1) reveals that in Latin American Spanish there is one language of communication, which means that both characters speak the same language: Latin American. Brian starts the conversation with a correct and grammatical structure: "Hola, mi nombre es Brian" (lit. Hello, my name is Brian). Since both characters can communicate in Latin American Spanish, they can understand each other, which implies that, unlike the previous discussion, there must be no confusion. Therefore, it is not clear why Brian then hesitates in his first and second utterances. Many reasons could justify his hesitation: Brian could have been nervous or afraid at the thought of not being understood (not linguistically, but maybe at what he wanted to ask), he may have expected the traveler to speak a different language, translators purposefully added hesitation as a direct transfer from American English audio/subtitling, etc. Whatever the reason, the Latin American dubbing not only ceases to maintain linguistic confusion and the metalinguistic conversation that appeared in the above-discussed cases, but it also changes the meaning of the conversation by introducing stereotypically cultural elements. Brian addresses the man and wonders if they can go with him. However, the man probably thought that Brian was being euphemistic because he answers, "pero déjeme decirte que te encontraste con la persona equivocada"/"yo no soy ningún pollero" (lit. "but let me tell you that you came across with the wrong person"/"I am not a smuggler").

In Spanish, "pollero" is a Mexican term that describes a guide for illegal immigrants to America (DRAE 2023). Since all characters are traveling in the direction of Austin, which is near the Mexico-United States border, Brian could have assumed that the man was a smuggler. However, as exemplified below, Brian's tone in the Latin American dubbing reveals surprise when the man states that he is not a trafficker, but an illegal. In this line, not only had Brian previously assumed the possibility of the traveler being a "pollero", but he expected him to be one: "¿No eres pollero?" (lit. "Aren't you a smuggler?"), "No, soy un simple ilegal" (lit. "No, I am just an illegal") and "Pero, ¿bromeas, cierto?" (lit. "You are kidding, right?"). This idea of surprise or even disbelief is ratified in the visual content, which displays Brian smiling (Figure 2). The relationship between the image and the American English audio, subtitling and Peninsular Spanish dubbing and subtitling is that Brian smiled with happiness (and maybe relief) at knowing that the traveler not only spoke the same language (American English in US DVD and Peninsular Spanish in SP DVD), but they could understand each other. In the Latin American dubbing, however, there is no reason for Brian wanting (or even needing) the man to be a smuggler. Brian leaves the carriage before entering Mexico and his goal was to find his mother and not to cross the border. This way, the relationship between the verbal and visual content is contradictory unless Brian momentarily digresses, and the focus of the conversation becomes the (im)possibility of the man being a "pollero".

The fact of dubbing the conversation around the issue of smuggling when Brian appears smiling is troublesome in two ways. First, the original meaning and intention of the conversation has disappeared. The characters no longer engage in a metalinguistic conversation and linguistic confusion in which L3$^{ST}$ and L3$^{TT}$ had the specific purpose of differentiating languages. As seen, L3$^{ST}$ was used to indicate the traveler's mother tongue (Latin American Spanish). In Peninsular Spanish, the introduction of another L3$^{TT}$ by means of "carro" (Latin American Spanish) not only marks the difference between two geographical varieties of Spanish but preserves the original L3$^{ST}$ meaning and function too. In Latin American dubbing, on the contrary, all conversation is carried out in Latin American (L2). As a result, L3$^{TT}$ disappears, the conversation acquires a completely different meaning, and it ultimately accentuates negative cultural stereotyping through the figure of "pollero". The other versions are not exempt from stereotypes since they start by

undermining the traveler's linguistic competence. However, (re)presenting the man as a smuggler pigeonholes him as the stereotypical version of Latin American immigrants in *Family Guy* (Medina-Vicent 2012).

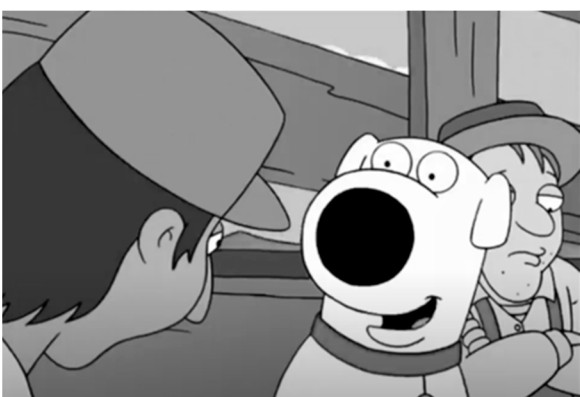

**Figure 2.** Brian smiling Source: SP DVD (2005) (Screenshot).

*3.2. Subtitling L3$^{ST}$: Latin American Spanish and Peninsular Spanish*

One of the main characteristics of subtitles is that they "must provide a semantically adequate account of the SL dialogue" (Díaz-Cintas 2010, p. 345). However, subtitles are not a readable version of dubbing material. In this line, "they must strive to capture the essence of what is said" (346). Regarding subtitling the specific scene of "Road to Rhode Island", Peninsular Spanish subtitling is the same as Peninsular Spanish dubbing (Tables 1 and 2). In fact, L3$^{ST}$ has been rendered in the same way, its original intention has been transferred and the meaning of the conversation has not been changed. Despite the similarities between Peninsular Spanish dubbing and subtitling, there is a relevant aspect regarding subtitling at the end of the conversation worth discussing. Although the traveler appears to be competent enough to correct Brian's performance in dubbing and subtitling, Peninsular Spanish subtitling reproduced the same grammatical mistake as in American English subtitling. Once Brian realized that the man spoke his mother tongue (English or Peninsular Spanish, respectively), he addressed him in his/their language.

Immediately, the traveler looked confused and appeared to no longer understand Brian. In the English audio, Brian responded with a "¿Qué?" ("What?") and in Peninsular Spanish dubbing as "What?". Both versions humorously debunk the traveler's apparent flawless bilingual skills. However, the fact that the man asks the question in the language that is supposed to be his native language brings further misunderstanding to the conversation. In English and Peninsular Spanish subtitling, the question is marked with intentionally wrong orthotypography. In this line, Peninsular Spanish subtitling adds double interrogation in English ("¿What?") while English subtitles eliminate one interrogation mark in Peninsular Spanish ("Qué?"). This way, subtitles not only indicate confusion between languages but also mark the man's lack of command in his non-native language, which continues to perpetuate the linguistic stereotypes associated with the migrant community (Medina-Vicent 2012).

As shown in the transcription of the Latin American subtitling (Table 2), some utterances have neither been subtitled nor indicate the language that is spoken. Similarly, the English subtitling in the US DVD only captioned Brian's performance in Spanish as "[Speaking Spanish]". Marking L3 by means of indicating the language that is spoken (Bartoll 2006), rather than the content, is a common strategy in subtitling L3. However, English audio (Table 1) confirms that Brian was speaking Latin American Spanish. Nevertheless, if the target audience does not understand the audible Latin American Spanish in the American DVD, American English subtitling does not compensate for it because it does not provide the content of the conversation that is spoken in Latin American Spanish. In this case, subtitles only indicate which language is used. The conversation in Latin

American subtitling is in Latin American Spanish (L2), which means that the non-subtitled parts correspond to English. However, the traveler corrects Brian's incorrect use of Spanish in "Pero cuando dices 'Me llamo es Brian'" (lit. "But when you say, 'I am called is Brian'"). This situation reveals that despite not being subtitled, Brian had been speaking in Latin American Spanish (L3$^{TT}$). Otherwise, the man would have corrected Brian's English. In this case, it seems that Brian mixed the structures "Me llamo Brian" with "Mi nombre es Brian" (lit. My name is Brian), which proves that no English was used.

When Brian realizes that he is being understood, he replies by saying, "Hablas inglés" (lit. "You speak English"). The fact that Brian mentions "English" corresponds to a case of signalization (Bleichenbacher 2008), and it indicates that L3$^{TT}$ was also English, which is ratified when the traveler says "Sólo eso. Y esto que estoy diciendo" (lit. Just this. And this that I am saying). In this sense, Latin American subtitling is confusing because it either implies that Brian speaks Latin American Spanish and English at the same time or that he is pretending to speak English when he is speaking in Spanish. If the target audience watched this scene in English (audio) with Latin American subtitles, they would listen to Brian speaking in Latin American Spanish, and they would logically assume that the non-subtitled part corresponded to Latin American until Brian admitted it was English, a small detail which may go unnoticed unless the audience pays attention. On the contrary, if the target audience listened to the Latin American dubbing with Latin American subtitling or even with American English subtitling, they would realize that the meaning of the conversation changed and that there is no connection between dubbing and subtitling. In this line, the Latin American dubbing is centered on intensifying the stereotypical representation of the migrant character by reserving the original meaning and intention of the conversation. English subtitling, however, maintains a difference between L1 and L3$^{ST}$ with the specific L3 role and function that establishes a metalinguistic conversation that concludes with confusion, while at the same time conveying linguistic and cultural stereotypes. As for the Latin American subtitling, the linguistic confusion does not derive from Brian and the man speaking different languages when they can understand each other but because the Latin American subtitling associates L3$^{TT}$ with both English and Latin American Spanish. This fact not only misleads the audience but also does not correlate in terms of content with Brian's first intervention and later correction.

## 4. Discussion

After a detailed analysis and discussion of the contexts of L1 and L3 in translation and the differences between the Latin American (US DVD) and Peninsular Spanish (SP DVD) dubbed and subtitled versions, this section explores the different techniques to represent L3 in subtitles (Bartoll 2006), L1 and L3 translation techniques (Martínez-Sierra et al. 2010), the role and function of L3 in translation (Corrius and Zabalbeascoa 2011; Corrius 2008) and the type of techniques used in translation (Molina and Hurtado 2002). First, Table 3 represents L2 and L3 combination in the Latin American and Peninsular Spanish dubbing and subtitling.

**Table 3.** L2 and L3 in Latin American and Peninsular Spanish dubbing and subtitling.

| "Road to Rhode Island" (American DVD, 2003) | | "El Camino a Rhode Island" (Spanish DVD, 2005) | |
| --- | --- | --- | --- |
| **Dubbing** | **Subtitling** | **Dubbing** | **Subtitling** |
| L2 > Latin American Spanish | L2 < Latin American Spanish<br>L3 < No subtitles | L2 < Peninsular Spanish<br>L3a < English<br>L3b < Latin American Spanish | L2 > Peninsular Spanish<br>L3a < English<br>L3b < Latin American Spanish |

As drawn from the table above, the main difference between Latin American (US DVD) and Peninsular Spanish (SP DVD) dubbing is the fact that in Latin American dubbing, not only does L3$^{TT}$ disappear but the difference and misunderstanding between languages is

replaced by a new context and meaning. In the case of Spanish dubbing, however, L1 and L3$^{ST}$ are reversed in order in translation, and L3b (i.e., Latin American Spanish) is added so to mark a difference between the two geographical varieties of Spanish (See Tables 4 and 5 for a complete analysis).

**Table 4.** L3 in Latin American Spanish dubbing: situation/operation, result/effect, and example.

| TT Situation/Operation | Result/Effect | Example |
|---|---|---|
| Situation: One language (Latin American Spanish)<br><br>Operation: Maintenance of Latin American Spanish in the whole TT conversation | • Elimination of L1 and L3$^{ST}$ difference in translation (L2 = Latin American Spanish and L3$^{TT}$ disappears)<br>• Replacement of the original meaning/function (metalinguistic conversation and linguistic confusion)<br>• Introduction of new and specific meaning in TT (cultural stereotyping) | "You speak English."<br>"Just that first speech.<br>And this one explaining it."<br><br>"Yo no soy ningún pollero."<br>"¿No eres pollero?"<br>"No, soy un simple ilegal." |

**Table 5.** L3 in Peninsular Spanish dubbing: situation/operation, result/effect, and example.

| TT Situation/Operation | Result/Effect | Example |
|---|---|---|
| Situation:<br>L2 ≠ L3 (+ another L3)<br>Operation 1:<br>Inversion of L1 (English) and L3$^{ST}$ (Peninsular Spanish) in TT | • L2 (Peninsular Spanish) ≠ L3$^{TT}$ (English)<br><br>L3 maintains the same function but reversed. | "Hola. Me llamo es Brian." (L3$^{ST}$)<br>"Hello. My name is de Brian." (L3$^{TT}$)<br>"Ah, let's see." (L1)<br>"Ah, veamos." (L2) |
| Operation 2:<br>• Substitution of L3$^{ST}$<br>Addition of different L3$^{TT}$ (signalization) | New L3$^{TT}$ (Latin American Spanish) marks the difference between the two varieties of Spanish. | "Señor, pare el auto!" (L3$^{ST}$)<br>"¡Señor, pare el *carro*!" (L3$^{TT}$) |

The situation between Latin American and Peninsular Spanish subtitling is the same if Peninsular Spanish subtitling and dubbing are compared (Table 3) but different between (1) the American and Spanish DVD and (2) the Latin American dubbing and subtitling. In the case of Peninsular Spanish, L1 and L3$^{ST}$ are reversed, and the subtitles add Latin American Spanish (L3b) in combination with L3a (English) to avoid L3 loss and change in function. If Peninsular Spanish subtitles and dubbing had not maintained English, a situation of linguistic similarity (L2 < Peninsular Spanish and L3$^{TT}$ < Latin American Spanish) would have resulted in confusion and very likely in L3$^{TT}$ loss: why would Brian change to a diatopic variation? Would not have the traveler understood Brian if he had spoken Peninsular Spanish?

In Latin American, L2 was employed in dubbing. Since the whole conversation occurred in such language, it was necessary to adapt the context and change the meaning of the situation. However, Latin American dubbing does not correspond to the context of dubbing since subtitling reproduces the context of misunderstanding. However, confusion (un)intentionally emerges when the utterances that are not subtitled are associated with two different L3s, as explained in the previous section and summarized below (Table 6):

**Table 6.** L3 in Latin American Spanish subtitling: situation/operation, result/effect, and example.

| Situation/Operation | Result/Effect | Example |
|---|---|---|
| Situation 1:<br>• L2: Latin American Spanish<br>• L3$^{TT}$: Latin American Spanish and English<br>Operation 1:<br>Repetition of L3$^{ST}$ (Latin American Spanish) | L2 = L3$^{TT}$ Standardization (L3 invisibility) | "Veamos." (L2)<br>"Pero cuando dices "Me llamo es Brian [ . . . ]"" (L3$^{TT}$) |
| Situation 2: No subtitles at the beginning<br>Operation 2: Mentioning of L1 (American English) in L3$^{TT}$ (signalization) | L3$^{TT}$ = English and Latin American Spanish Contradiction: L3 is associated with two languages | "Pero cuando dices "Me llamo es Brian [ . . . ]"" (L3$^{TT}$)<br>"Hablas inglés." (L3$^{TT}$) |

Regarding the techniques and the role and function of L3 in translation (Zabalbeascoa 2018; Corrius and Zabalbeascoa 2011; Corrius 2008), how L3 is represented (Bartoll 2006), L1 and L3 translation techniques (Martínez-Sierra et al. 2010) and which techniques are used in translation (Molina and Hurtado 2002), there are some differences between versions (US and SP DVD), languages (Peninsular Spanish and Latin American Spanish) and format (dubbing and subtitling). Regarding the role and function of L3 in the US DVD, dubbing has eliminated any possible conflict or tension between L2 and L3 because the conversation is conducted in just one language. In this sense, L3 has disappeared, and the meaning of the situation has changed. In Latin American Spanish, signalization (Bleichenbacher 2008) is applied but L3 has been wrongly associated with two languages, which has created confusion. Since it has been demonstrated that one of the L3s could not be English (even if subtitling mentions it), L3 must correspond to Latin American. In this case, however, L3$^{TT}$ is imperceptible because it has not been compensated (Corrius and Zabalbeascoa 2011). As for both subtitling and dubbing in the Spanish DVD, Brian's L2 is Peninsular Spanish but he makes use of two L3s: English (L3a$^{TT}$) to address the traveler and Latin American Spanish (L3b$^{TT}$) to order the driver to stop. This is a case of L3$^{ST}$ = L2, since in the original subtitles L3 was Peninsular Spanish. Languages were reversed (L1 < English and L3$^{ST}$ < Peninsular Spanish become L2 < Peninsular Spanish and L3$^{TT}$ < English). However, to avoid Latin American neutralization and loss, both dubbing and subtitling include another L3. Despite its similarities to L2, L3$^{ST}$ function and role is kept in translation.

In Latin American subtitling, L3 is not marked. However, subtitles indicate that the traveler was speaking in English: "hablas inglés". In the US DVD, English subtitles indicate that Brian was "[Speaking Spanish]", which is a technique that marks L3 by indicating the language in brackets (Bartoll 2006). In the case of Peninsular Spanish, L2 becomes Peninsular Spanish and as for L3$^{TT}$ two languages are included: American English and Latin American Spanish. In both cases, they differ from L2. In this version, both L3s are marked in italics (Bartoll 2006) to differentiate them from L2. The use of italics, however, is different among languages: Peninsular Spanish employs italicization when L3s are used, Latin American does not italicize, and nor does English subtitling (SP DVD). In the American DVD, however, italics are used to mark the ungrammatical utterances: *"Me llamo es Brian"*, *"es"* and *"Qué?"*.

In their analysis, Martínez-Sierra et al. (2010) associate self-translation with an immigrant character that speaks both "his or her native language and also Spanish" (25). In the episode analyzed, self-translation occurs in both the English audio and Peninsular Spanish dubbing, for the traveler is competent in both languages and he could perfectly be native in both. Even though he does not "facilitate communication with Spanish characters" (25), he does it with non-native speakers of Spanish, such as Brian. In these two cases, the conversation occurs in both English and Spanish. In the contexts of both the American and the Spanish DVD, there is no need to translate the parts that are spoken in Spanish (US DVD) and English (SP DVD) because (1) the main objective is to keep L1 and L3 difference in translation, (2) the DVD versions provide subtitles and (3) the utterances can

be understood easily by the audience. In this regard, the fact that an immigrant character is proficient in both languages empathizes "with both the other Spanish characters in the film and the audience" (25).

As for the translation techniques that have been used in subtitling and dubbing, there are some differences worth commenting on. In Latin American subtitling, there is a clear omission of subtitles. Since the traveler corrects Brian's "Me llamo es Brian", one can imagine that this is the structure that he said (from English "My name *is* Brian"). As it has not been subtitled, the inclusion of the language that Brian might have spoken ("inglés") could work as a compensating element (Molina and Hurtado 2002). In Latin American dubbing, the start and end of the conversation corresponds to the original version. When it comes to the conversation, the meaning has been changed and adapted culturally, despite creating stereotypes towards the Latin American community. In the case of SP DVD, the situation is the same between subtitling and dubbing: the first two interventions are literal translations (Molina and Hurtado 2002) since the structure is the same as in the English subtitles (the only changes applied are to mark ungrammaticality). However, when Brian realizes that both are speaking the same "idioma" and the traveler admits having been "doblado", Spanish subtitles make use of generalization (in the case of "language", which avoids confusion by not being precise about which language they speak) and description, in the case of the physical position of the character.

## 5. Conclusions

Multilingualism (or plurilingualism) is a key element in *Family Guy*, which represents otherness through the inclusion and mixture of characters whose main language of communication (L1) is not English but others (L3). The use of L3 in the sitcom, however, is mainly centered to stereotype characters and cultures. The main goal of this paper was to study multilingualism and multiculturalism in *Family Guy* in the analysis of the role and function of L3$^{TT}$ (Zabalbeascoa 2018; Corrius and Zabalbeascoa 2011; Corrius 2008) to check if linguistic and cultural stereotyping was portrayed in translation. To do so, a comparison between dubbing and subtitling from the American and Spanish DVD versions of the episode "Road to Rhode Island" was carried out. In this regard, the analysis took into consideration L1 and L3 in translation into two geographical varieties of Spanish: Peninsular Spanish (SP DVD) and Latin American Spanish (US DVD). The paper incorporated a thorough analysis of L1 and L3 translation techniques (Martínez-Sierra et al. 2010), L3 representation or marking in subtitles (Bartoll 2006), and the specific translation techniques (Molina and Hurtado 2002) that were used in translation.

Despite the shortness of the scene that was analyzed, the study of Latin American Spanish and Peninsular Spanish dubbing and subtitling revealed some interesting differences in L3$^{TT}$ role and function, L3$^{TT}$ meaning, and techniques employed in translation. As shown during the analysis, Peninsular Spanish dubbing and subtitling were the same. In subtitling, English (US DVD) marked the use of L3 by indicating in brackets the language spoken (Bartoll 2006), as in "[Speaking Spanish]", and Peninsular Spanish marked it in italics (Bartoll 2006). In English subtitling (US DVD), however, italics were employed to mark ungrammaticality. The main difference when compared to Latin American Spanish and English audio and subtitling is the fact that Peninsular Spanish dubbing and subtitling are aimed at keeping L3 in translation. In this case, L1 and L3 languages were reversed. Therefore, L3 was not lost, but compensated (Zabalbeascoa 2018; Corrius and Zabalbeascoa 2011; Corrius 2008) by adding a new L3. This is the only case that included three languages in translation: L2 (Peninsular Spanish), L3a (English), and L3b (Latin American Spanish) with the use of "carro". Moreover, signalization (Bleichenbacher 2008) appeared in English audio and subtitling ("you speak English"), Latin American subtitling ("hablas inglés"), and Peninsular Spanish dubbing and subtitling (the hypernym "habla mi idioma"). Moreover, the technique of self-translation (Martínez-Sierra et al. 2010) was useful to explain the native nature of the immigrant character and debunk the false idea of foreigners not being competent in English (Medina-Vicent 2012).

One of the greatest differences was the cultural adaptation of Latin American Spanish subtitling. The whole context and meaning of the situation were changed and the addition of "pollero" included stereotyping towards the Latin American community. In the case of Peninsular Spanish dubbing and subtitling, the metalinguistic confusion continued, even with the introduction of the pun "doblado" (i.e., "bent" and "dubbed") which aligned with the verbal and visual context while it added a humorous touch that harmonized perfectly with Brian's surprise: "¿Me toma el pelo?". Finally, the analysis demonstrated a variety of techniques in translation, both in subtitling and dubbing, i.e., omission of subtitles in Latin American, compensation (Molina and Hurtado 2002) to specify the language, cultural adaptation and change in meaning in Latin American dubbing, literal translations in Peninsular Spanish, generalization to avoid language precision ("idioma") and description for "doblado".

Overall, the study of multilingualism and multiculturalism in *Family Guy* through L1 and L3 and/in translation in a short scene of the episode "Road to Rhode Island" has been useful to give visibility to a nameless and immigrant character that would have otherwise been considered part of the great amalgam of characters that conform the Latin American community in *Family Guy*. Regarding L3 dubbing and subtitling, the analysis has demonstrated that Latin American Spanish dubbing and subtitling did not consider preserving the L3 role and function in translation. The omission of subtitles created confusion and an erroneous association of two languages in L3$^{TT}$. In dubbing, a complete change in the meaning of the conversation purposefully (or not) introduced cultural stereotypes. However, it was the Spanish dubbing and subtitling that not only avoided cultural stereotyping but also preserved the original L3 role and function by combining two varieties of Spanish to remark language variety and diversity and give visibility to L3 (or rather, L3s) in translation.

**Funding:** This research received no external funding.

**Institutional Review Board Statement:** Not applicable.

**Informed Consent Statement:** Not applicable.

**Data Availability Statement:** The data that support the findings of this study are available from the corresponding author upon reasonable request.

**Acknowledgments:** My heartfelt thanks to Eva Espasa Borràs and Patrick Zabalbeascoa for their support and guidance.

**Conflicts of Interest:** The authors declare no conflict of interest.

## Notes

[1] 'The Former Life of Brian' (Season 6 Episode 9).

[2] *Family Guy* was canceled on two occasions. When season 1 (1999) was broadcasted, it included 7 episodes. Season 2 (1999–2000) was broadcast with 21 episodes. This distribution is kept in the American DVD. In the Spanish DVD, season 1 contains 14 episodes and season 2 includes 15 episodes. Due to this different reorganization, some episodes do not coincide between the SP and US DVDs. For example, the episode 'Da Boom' (season 2) is the first on the Spanish DVD but the third on the American DVD.

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
