# Peer review of "Multilingualism and Multiculturalism in Family Guy: Challenges in Dubbing and Subtitling L3 Varieties of Spanish"

_languages, doi:10.3390/languages8020143_

Round 1

Reviewer 1 Report

The manuscript clearly lays out the aims/objectives of the study, the previous literature and research gaps, and its contribution to the field. The selected scene provides a clear analysis about the linguistic choices and differences.  

Author Response

Dear Sir/Madam,

First of all, thank you for your comments. I do appreciate the feedback I got. Despite the fact that I was not given specific advice to modify parts of the manuscript, I did a thorough revision and corrected some linguistic aspects (i.e., grammar and spell check).

Yours faithfully,

Reviewer 2 Report

This is an in-depth analysis of a short video passage of the TV series Family Guy. The author offers an interesting discussion under the area of translation related to the ways a few characters of the series communicate in a different language from the original English and also about the choices made in presenting the dubbing and subtitling in two Spanish varieties. The discussion is well-conducted and supported by frequent use of the previous investigations/literature in the field. The use of the material and the method employed are appropriate for this type of research and the comparisons of the data are shown in a clear way. The discussion and the results prove the effects of the translations choices into cultural tensions in the language contact situation in the United States. The manuscript is well written, and the reader can follow it without major problems despite the fact that multiple elements are being considered in the discussion regarding the L1, L2, and L3.

In my opinion, there is one important aspect that needs to be clarified. It would be beneficial to incorporate a discussion about the L3 (“a third language that refers to any other language(s) in the text," p. 2) in the introduction. One or two examples and their explanation would help the reader to understand this concept from the beginning and follow the discussion without getting confused.

Author Response

Dear Sir/Madam,

First of all, I would like to thank you for your comments and suggestions, since they helped me to better the manuscript. In this regard, I do appreciate all the feedback I got.

One of your comments was related to the mentioning of L3 in the introduction. I was suggested to add some discussion prior to the analysis and results in order to clarify the concept and avoid later confusion, since I discuss L1, L2, and L3 (ST and TT) quite often during the study.

  • Right after the definition of L3 (Corrius and Zabalbeascoa (2011) and Zabalbeascoa (2018)) on page 2, I have showed L1, L2 and L3 in context, that is, how they all work in Family Guy. In this regard, I exemplified how these languages can be found in the series and the reason why L3 needs to be different from L1 and L2, following Zabalbeascoa (2018).
  • In applying this change, which I later exemplify in detail when it comes to the episode I analyse (i.e., I state which languages represent L1, L2 and L3 in the episode), I clarified the concept in relation to the series so it is not confusing.
  • In a similar line, I revised the whole manuscript and paid careful attention to the concepts that might have brought confusion if not explained in detail, or cases in which they were not clearly laid out. In this regard, I made them explicit (for example, on page two I clarified the concepts of ST and TT and I made sure their use was coherent in all the study) and I specified whether I was referring to ST or TT when discussing L3, since there were some cases in which its distinction was not clear enough, such as in section 4 (Discussion).
  • Finally, I revised and confirmed that all discussions around L3 were clear throughout the text. For example, in section 4. Discussion (the paragraph below Table 5 and the paragraph following Table 6).

Another aspect that I was indicated to improve was related to research design, questions, hypotheses and methods. This study focuses on multilingualism and multiculturalism in Family Guy, which is analysed through the role and function of L3 in translation and accompanied by an analysis and discussion on the techniques to represent L3 in traslation, L1 and L3 techniques and general translation techniques employed. In doing so, my goal is to check whether subtiling and dubbing portray stereotypes in translation and ultimately give voice to an unknown character.

  • In order to clarify all the aims, I redid part of the final paragraf in the introduction (page 3, middle-end of the paragraph) by listing and numbering all aims, from general to specific.
  • In also modified part of section 2. Materials and Methods, where I verified that the aims I have were properly linked to my theoretical framework. This is the reason why there have been some changes in this section.
  • Finally, I revised the conclusion, so to check all objectives were present, clear and well discussed, again, from general to specific.

Other changes that I have applied:

  • I did a thorough revision of the text and corrected different linguistic aspects (grammar and spell check).

Yours faithfully,